# Barriers and Facilitators to the Implementation of Effective Alcohol Control Policies: A Scoping Review

**DOI:** 10.3390/ijerph19116742

**Published:** 2022-05-31

**Authors:** Jintana Jankhotkaew, Sally Casswell, Taisia Huckle, Surasak Chaiyasong, Payao Phonsuk

**Affiliations:** 1Social and Health Outcomes Research and Evaluation & Whariki Research Centre, Massey University, Victoria Street West, P.O. Box 6137, Auckland 1142, New Zealand; s.casswell@massey.ac.nz (S.C.); t.huckle@massey.ac.nz (T.H.); 2International Health Policy Program, Ministry of Public Health, Tiwanon Road, Nonthaburi 11000, Thailand; surasak.c@msu.ac.th; 3Alcohol and Health Promotion Policy Research Unit and Social Pharmacy Research Unit, Faculty of Pharmacy, Mahasarakham University, Kham Riang, Kantharawichai District, Maha Sarakham 44150, Thailand; 4Department of Health Education and Behavioral Sciences, Faculty of Public Health, Mahidol University, Ratchawithi Road, Ratchathewi District, Bangkok 10400, Thailand; payao.pho@mahidol.ac.th

**Keywords:** policy implementation, effective alcohol policies, barriers, facilitators

## Abstract

Implementation of effective alcohol control policies is a global priority. However, at the global and national levels, implementing effective policies is still challenging, as it requires commitment from multiple stakeholders. This review provides a synthesis of barriers and facilitators to implementing effective alcohol control policies. We conducted a scoping review from two main databases: Scopus and Web of Science, and the grey literature from the World Health Organization’s website. We included any studies investigating barriers and facilitators to implementing four effective policies: Alcohol pricing and taxation, control of physical availability, alcohol marketing control, and drink-driving policy. Articles published between 2000 and 2021 were included. The search yielded 11,651 articles, which were reduced to 21 after the assessment of eligibility criteria. We found five main barriers: resource constraint; legal loopholes; lack of evidence to support policy implementation, particularly local evidence; low priority of policy implementation among responsible agencies; and insufficient skills of implementers. Facilitators, which were scarce, included establishing monitoring systems and local evidence to support policy implementation and early engagement of implementing agencies and communities. We recommend that national governments pay more attention to potential barriers and facilitators while designing alcohol control regulations and implementing effective policies.

## 1. Introduction

Implementation of an alcohol control policy is a global priority. Alcohol contributes to more than 230 health conditions and has a negative impact on both individuals and society [1]. Reducing alcohol consumption is a global commitment; for example, reducing alcohol consumption is one of the Sustainable Development Goals of the United Nations, and the World Health Organization (WHO) adopted the Global Strategy to Reduce Harmful Use of alcohol (Global Strategy) in 2010. However, the implementation of effective policies is still a global and national challenge. WHO’s report reviewed the progress of the Global Strategy and addressed the challenges regarding the implementation of the Global Strategy at global and national levels over the past decade. The report reiterates the challenges related to policy implementation of effective policies, including lack of political commitment, limited technical capacity, human resources and funding [2].

Implementation of an alcohol control policy is complex, as it often requires efforts from various stakeholders beyond health sectors, protection from various vested interests and is influenced by social and cultural factors within organisations and society in general [2]. Implementing an effective policy requires individuals, organisations, and systems with enough capacity to enable policy implementation. Individuals and organizations require technical, administration, and political skills to effectively implement alcohol control policies. The systems’ capacities are the environments that help to accelerate implementation, for example, political commitment and social climate promote policy implementation [3]. Barriers and facilitators are embedded in those policy capacities. Removing barriers and promoting facilitators requires a systematic synthesis that can map and analyse how best to help governments design an effective policy and develop systems that can foster implementation. In addition, there are potential factors to consider for implementing general health-related policies, including characteristics of interventions or policies, factors inside implementing agencies, factors outside the control of implementing agencies, and individuals involved in policy implementation [4].

The main contribution of this study to the existing literature is to undertake a scoping review of barriers and facilitators to policy implementation of the effective alcohol control policies (i.e., alcohol pricing and taxation, control of physical availability, control of alcohol advertisement, drink-driving policy). There is no existing systematic scoping review (or systematic review) on barriers and facilitators to implementing the effective alcohol control policies. Only one systematic review has previously provided a synthesis of barriers and facilitators of alcohol control policy implementation but focused only on the screening and brief intervention of alcohol use [5]. The implementation of screening and brief interventions has occurred in health care settings, while other effective policies such as taxation and pricing policy, alcohol marketing control, control of physical availability, and drink-driving policy are implemented in different settings and involve more stakeholders. Therefore, barriers and facilitators to policy implementation may vary depending on the context and settings of the policy. To systematically scope barriers and facilitators to policy implementation from various settings can help countries be informed and design effective implementation of the effective alcohol control policies.

This review provides a synthesis of barriers and facilitators to implementing effective alcohol control policies.

## 2. Materials and Methods

We conducted a scoping review using the Joanna Briggs Institute (JBI) guidelines [6] and registered the protocol of the scoping review at the Open Science Framework. The main research question of this review is “what are the barriers and facilitators to implementation of effective alcohol control policies?”. Our review focused on effective regulatory policies, including alcohol taxation and pricing, control of marketing (i.e., alcohol advertisement, promotion, pricing promotion, alcohol sponsorship, products, and placement [7]), control of physical availability (i.e., regulating retail outlets, the density distribution of retail outlets, restricting hours and days of trade, ban on public drinking, minimum purchasing age, licensing, control of social supply, and online sales [7]), and drink-driving measures.

In this review, policy implementation included carrying out, accomplishing, fulfilling, producing, and completing policy goals [8].

### 2.1. Search Strategy and Selected Databases

We developed a search strategy in Scopus and revised it appropriately for the Web of Science and WHO’s website. We chose Scopus because it is the largest search engine in the scientific field [9], including 100% of MEDLINE health science topics, and we selected Web of Science, as some of its articles are not covered in Scopus. We also included grey literature from WHO’s website. The key search strategy is provided in the Appendix A. We conducted the search on 18 May 2021.

### 2.2. Inclusion Criteria and Exclusion Criteria

We included studies that addressed barriers and facilitators of the four effective policies from literature published worldwide between 2000 and 2021 (we did not include brief intervention as this has previously been reviewed [5] and is not a population-wide prevention approach). We included studies that investigated factors influencing the alcohol control policy implementation processes and outcomes, even if the studies did not explicitly mention barriers or facilitators to the alcohol control policy implementation. We included both published and grey literature (i.e., technical reports from WHO’s website), and studies that applied any study design, including qualitative, quantitative and mixed-methods.

We excluded studies that did not address barriers and facilitators to the implementation of the four effective policies stated above, studies that were not in English, and studies that did not provide details on the methods (e.g., editorials, debates, news) to ensure transparency of studies (e.g., methods, study design and data collection).

### 2.3. Evidence Screening, Selection, Data Charting, and Data Analysis

Titles and abstracts were independently screened by two reviewers (JJ and PP) following the review protocol. The full texts of studies were later selected and assessed for eligibility criteria. There was no disagreement between the two reviewers for screening titles and abstract and full text screening.

The template of data charting from JBI was adapted [10] during the protocol setting stage and piloted and adjusted during the review stage. We designed data charting according to the research question and objective. The data charting form included authors, country, study design, study population, and barriers and facilitators to policy implementation. Barriers are factors that delay or have negative effects on policy implementation, and facilitators are factors that positively influence or enable policy implementation.

Prior to use, we tested the data charting form and discussed improvements to its comprehensiveness and clarity. One reviewer (JJ) charted the data, and the other (PP) verified its accuracy.

To analyse the data, we categorised information into barriers and facilitators. We applied the Preferred Reporting Items for Systematic Review and Meta-Analysis extension for Scoping Review (PRISMA-ScR) checklist (see Appendix A).

## 3. Results

### 3.1. Search Results

We identified 11,651 articles. We did not find any literature from WHO related to barriers and facilitators to policy implementation. After removing duplication, 8189 articles remained. After excluding papers not in the scope of this review, twenty-one were assessed for eligibility, and we included all of these studies in the synthesis (Figure 1).

### 3.2. Characteristics of Selected Studies

The majority of studies were conducted in high-income countries and applied a qualitative approach. Most studies were conducted among implementing agencies and implementers who were involved in policy implementation. Few studies were conducted among target populations of policies (e.g., alcohol retailers) (Table 1).

### 3.3. Barriers to the Implementation of the Four Policies

Within 19 of the 21 included studies, five main barriers to policy implementation were identified. These included resource constraints, legal loopholes and complications of law, insufficient evidence and lack of monitoring systems to support policy implementation, a low priority among responsible authorities and decision-makers, and limited capacity of implementers and implementing agencies. First, ten out of nineteen studies found resource constraints, such as materials, human resources, and the high workload of police officers, were barriers [11,17,18,23,26,27,28,29,30,31].

Second, six studies reported that legal loopholes and legal complications (e.g., requiring precise law interpretation in practice, unclear roles of responsible authorities in legislation) can be a bottleneck for policy implementation [11,15,20,26,28,30]. For example, the study in Thailand addressed legal loopholes in alcohol marketing control; advertisement of alcohol products is prohibited, but not non-alcoholic products. The alcohol industry seized this loophole to promote alcohol brands by using non-alcoholic products [11]. Another example related to the interpretation of the law from different stakeholders is found in the study conducted in England; the government introduced cumulative impact policies, which provides power to local authorities to grant or not grant alcohol licenses by considering the impact of alcohol licenses to areas. However, the cumulative impact policies’ interpretation differed across different involved stakeholders (e.g., local residents, and licensing applicants) [15]. Another example of legal complications is found in Australia, where the role of implementing agencies (i.e., licensing authorities and police) are not clearly stated in the legislation. This caused the reluctance of responsible agencies (i.e., police) to implement the policy [20].

Third, six studies addressed a lack of evidence [13,16,21,31] and monitoring systems [11,18] to support alcohol policies. Most referred to the need for local evidence for supporting decisions in the licensing application process. For example, studies in England and Scotland found insufficient robust evidence to support a decision on the defence of new licenses [13,16].

Fourth, five studies addressed a low priority of policy implementation among responsible authorities and decision-makers [13,25,27,28,31]. Among these studies, the low value placed on alcohol-related problems by agencies responsible for policy implementation was addressed [13,28]. The study in Scotland also addressed a low priority of public health interest among implementing agencies, but rather they focused on economic development instead [13]. Another aspect is that implementers, police officers, for example, did not believe in the effectiveness of random breath testing [27]. Therefore, they tended to place a low priority on implementing the policy.

Finally, four studies reported that it required implementers’ skills to implement in practice, but they had insufficient knowledge about law and skills to implement the policy effectively [16,17,19,20]. For example, a study in London found that councillors acted as the chair of licensing board who decided to grant alcohol licenses. However, the study reported that they had limited knowledge about licensing matters and law. The study also reported that the trainings given were insufficient to perform the chair of licensing board’s functions. Sometimes, the licensing board had to draw upon a legal team to support the implementation processes [16].

Some of the barriers to policy implementation were commonly found across all policies, but others were related more specifically to types of policies. For example, resource constraints, legal loopholes, and low priority of implementing agencies and policymakers were found in all policies. Whereas, lack of local evidence to support policy implementation and insufficient skill of implementers were mainly found in relation to the physical availability control, particularly licensing policy and enforcement of a minimum purchasing age.

Apart from the five main barriers, few studies addressed other factors that hindered policy implementation, including drinking norms, conflicting interests, and the alcohol industry’s role in promoting drinking norms. Two studies addressed drinking norms, which are the acceptability of alcohol drinking in daily life among people in society, and hindered implementation [11,28]. In addition, the studies in Nigeria and Thailand illustrated that the alcohol industry attempted to promote drinking as a custom and tradition in daily life [11,28]. This can cause difficulty and reluctance of governments to commit to and implement alcohol control policies. Another four studies illustrated conflicting interests among different sectors [15,16,30,31]. For example, alcohol retailers and local authorities viewed alcohol sales particularly as a part of the night-time economy and as a source of income in communities [16]. The economic sector has its main purpose of generating revenue; therefore, alcohol control and public health interest may not be included in their main agenda [30]. 

### 3.4. Facilitators to the Implementation of the Four Policies

Nine studies addressed facilitators to policy implementation. Four studies stated that having evidence and a monitoring system to support policy implementation was key to achieving implementation outcomes [12,13,15,20]. One study addressed the engagement of key stakeholders at an early stage of legislative processes [20], and another study stated the importance of community engagement that helped accelerate the implementation [16]. One study reported public support as a key facilitator to implementation [24]. Other key facilitators addressed by included studies are the collaboration between implementing agencies [26] and motivation of implementing agencies [18].

## 4. Discussion

We found five common barriers to policy implementation: (1) resource constraints, (2) legal loopholes and legal complications, (3) insufficient evidence and lack of monitoring systems, (4) a low priority of policy implementation among responsible agencies, and (5) limited capacity of implementers and implementing agencies. Apart from these five barriers, some studies addressed external barriers such as the alcohol industry promoting drinking as a norm and conflicting interests among different sectors.

The majority of studies address barriers within the implementing agencies, for example, resource constraints and a lack of monitoring systems. Some of the findings in this review were similar to the systematic review on barriers and facilitators to implementing screening and brief intervention for alcohol misuse [5]. Johnson et al. (2011) stated that the main barriers to effective implementation were a lack of resources, training, support from management, and excessive workloads. However, we found additional and important aspects: legal loopholes and legal complications (e.g., requiring precise legal interpretation in practice, unclear roles of responsible authorities in legislation). Legal loopholes created room for the alcohol industry to seize the opportunity to promote its benefits [11]. Therefore, a comprehensive alcohol control law is required at the design stage [32], and countries need to ensure update-to-date alcohol control law regarding social and cultural context as well as adaptation of the alcohol industry’s strategies [33]. Apart from that, the interpretation of alcohol control laws requires legal support from legal experts [16].

Another aspect relevant to regulation was the low priority given to policy implementation among responsible agencies and decision-makers. In the implementation of regulatory measures, governments often apply a “top-down” approach [34]. Policy formulation and policy implementation in many situations are the responsibility of different actors [34]. Because of that, implementing agencies and implementers are not involved in policy formulation [34,35]. Therefore, the implementation of the effective policies may not be a main priority of implementing agencies. Furthermore, policy implementation is not politically attractive, and policy formulation is seen as more important than policy implementation [34]. Another factor is the drinking norms in everyday life, which might result in the reluctance of implementing agencies to address the problems [11,28]. These issues may explain why there is a low priority for policy implementation among responsible agencies and decision-makers.

Lack of knowledge about law content and skills for policy implementation was one of the prominent barriers to policy implementation. There are various explanations for insufficient knowledge of law content and policy implementation skills. First, law enforcement requires specific skills and knowledge (e.g., legal content) [16]. Therefore, legal support and specific training are needed to effectively implement effective policies. Another factor is the lack of resources for training and the high workload of implementers, and this could result in insufficient knowledge and skills. Improving the knowledge and skills of individuals (in this case, implementers) requires investment from organisations and systems [3]. If central governments did not allocate a sufficient budget to an organisation, a deficit of skills could occur.

Another barrier to effective policy implementation is conflicting goals of public health and economic interest among different sectors. This barrier occurs because effective alcohol control policy implementation requires cooperation with multiple stakeholders from various interests, including government sectors, alcohol retailers, and the alcohol industry [2]. Various strategies can be applied to handle conflicting goals of public health and economic interest. For example, to ensure policy priority across various sectors, multisectoral collaboration at the policy formulation stage is required [36]; however, the participatory process should be conducted free of conflicts of interest, especially from the alcohol industry. Apart from that, with various interests from different sectors, competent coordinating organisations with legitimacy and strong ownership are required to promote effective coordination across different sectors [3]. More importantly, based on the findings of this review, the alcohol industry negatively influenced policy implementation by promoting drinking norms and creating resistance to policy implementation [11,28]. Therefore, a comprehensive regulation to regulate industry strategies and roles in the policy process, including the implementation process, is urgently needed to promote the implementation of effective alcohol control policies.

### Limitation

The limitation of this scoping review is concerned with restricting the literature search to articles published in English, resulting in some potentially relevant studies being omitted.

## 5. Conclusions

This review highlighted five main barriers to policy implementation, including insufficient resources, exploitation of legal loopholes, a lack of monitoring systems and local evidence to support policy implementation, a low priority for implementation by responsible agencies, and a lack of skills among implementers. Facilitators of policy implementation were sparse; they included monitoring systems and local evidence to support policy implementation and early engagement of implementing agencies and communities. We recommend that governments should allocate more resources (financial, material, and human) to support more effective policy implementation and provide sufficient training for implementers. To design effective policies and ensure better implementation in practice, comprehensive policies and clear guidelines, as well as public communications to promote public acceptance, can help to effectively implement and achieve policy goals.

## Figures and Tables

**Figure 1 ijerph-19-06742-f001:**
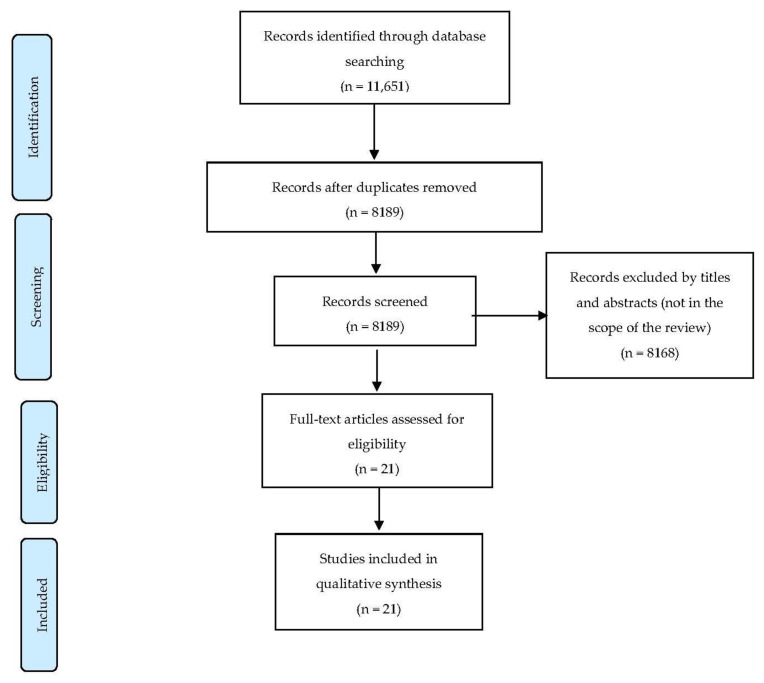
Article screening process.

**Table 1 ijerph-19-06742-t001:** Barriers and facilitators to policy implementation among the four policies.

	Author	Year	Title	Country	Method and Target Population of Studies	Barriers and Facilitators to Policy Implementation	Refs.
	Alcohol marketing						
1.	Kaewpramkusol, R., Senior, K., Nanthamongkolchai, S., & Chenhall, R.	2019	A qualitative exploration of the Thai alcohol policy in regulating alcohol industry’s marketing strategies and commercial activities	Thailand	Qualitative study Target population: implementers and relevant stakeholders (government officers, academia, and civil society)	Barriers: acceptance of drinking alcohol as a social norm, adaptation of alcohol marketing strategies to take advantage of a legal loophole, legal loopholes in Thai alcohol control law on alcohol marketing, lack of monitoring of digital marketing, high workload, and limited resources for enforcement, alcohol industry’s strategy to promote alcohol as an ordinary product and a part of everyday socialising, alcohol sponsorship provides economic and social benefits resulting in acceptance of the alcohol industry sponsorships	[11]
	Physical availability						
2.	Egan, M., Brennan, A., Buykx, P., De Vocht, F., Gavens, L., Grace, D., Halliday, E., Hickman, M., Holt, V., Mooney, J.D., Lock, K.	2016	Local policies to tackle a national problem: Comparative qualitative case studies of an English local authority alcohol availability intervention	England	Qualitative study Target population: implementers and relevant stakeholders (i.e., local government authority, licensing leads, public health, police, and other key stakeholders)	Facilitators: using local evidence-informed decision making and skills of licensing leaders in negotiation	[12]
3.	Fitzgerald, N., Nicholls, J., Winterbottom, J., & Katikireddi, S. V.	2017	Implementing a public health objective for alcohol premises licensing in Scotland: A qualitative study of strategies, values, and perceptions of evidence	Scotland	Qualitative study Target population: implementers (i.e., public health officers involved in implementing licensing policies and working with the Licensing Board, main implementing agencies)	Barriers: lack of priority among key implementing agencies, and licensing board (i.e., the decision-making body for issuing alcohol licenses), implementing agencies make little use of evidence Facilitators: important available evidence/data on alcohol-related harms to support decision making	[13]
4.	Gosselt, J. F., Van Hoof, J. J., & De Jong, M. D. T.	2012	Why should I comply? Sellers’ accounts for (non-) compliance with legal age limits for alcohol sales	Netherlands	Qualitative study Target population: target groups of policy (i.e., managers or owners of alcohol outlets)	Barriers: lack of knowledge of law among alcohol sellers, inability of staff to manage aggression at alcohol retailers, lack of motivation of alcohol sellers to comply with laws Facilitators: motivation of alcohol retailers, and knowledge of the law among alcohol retailers	[14]
5.	Grace, D., Egan, M., & Lock, K.	2016	Examining local processes when applying a cumulative impact policy to address harms of alcohol outlet density	England	Qualitative study Target population: implementers (i.e., licensing officers, councillors, police, and trade)	Barriers: interpretation of regulation differed across local authorities (legal loopholes), and economic benefits outweigh public health consideration among implementers/alcohol retailers Facilitators: evidence-based decision making	[15]
6.	Herring, R., Thom, B., Foster, J., Franey, C., & Salazar, C.	2008	Local responses to the Alcohol Licensing Act 2003: The case of Greater London	England	Qualitative study Target population: implementers (i.e., licensing officers and chairs of licensing committees)	Barriers: legal challenges, insufficient robust evidence, inadequate data, lack of training among councillors, lack of support from decision-makers, and a balance between economic versus public health benefits Facilitators: engagement of residents	[16]
7.	Miller, P. G., Curtis, A., Graham, K., Kypri, K., Hudson, K., & Chikritzhs, T.	2020	Understanding risk-based licensing schemes for alcohol outlets: A key informant perspective	Multi-country: Canada and Australia	Qualitative study Target population: implementers	Barriers: lack of knowledge of law among police officers and limited resources	[17]
8.	Mooney, J. D., Holmes, J., Gavens, L., De Vocht, F., Hickman, M., Lock, K., & Brennan, A	2017	Investigating local policy drivers for alcohol harm prevention: A comparative case study of two local authorities in England	England	Qualitative study Target population: implementers (i.e., police, public health, commissioning, treatment service/clinical, information analyst, and licensing/trading standard)	Barriers: resource constraints, and information sharing was difficult-information technology compatibility issues between implementing agencies (e.g., alcohol-related harm data) Facilitators: pro-active police with strong motivation to tackle the poor police image of the city in relation to drinking and licensing	[18]
9.	Puangsuwan, A., Phakdeesettakun, K., Thamarangsi, T., & Chaiyasong, S	2012	Compliance of off-premise alcohol retailers with the minimum purchase age law	Thailand	Mixed-methods Target population: target of policy (i.e., alcohol retailers)	Barriers: lack of knowledge of the law	[19]
10.	Trifonoff, A., Nicholas, R., Roche, A. M., Steenson, T., & Andrew, R.	2014	What police want from liquor licensing legislation: the Australian perspective	Australia	Qualitative study Target population: implementers (i.e., police officers)	Barriers: unclear roles of authorities in implementation, influence of alcohol industry in decision-making, inability of police to prove intoxicated persons, and disconnection between decision-makers and implementors Facilitators: involvement of police in legislative and regulatory processes, partnerships (including key stakeholders as a partnership in implementation), and using data for action and decision making	[20]
11.	Wilkinson, C., MacLean, S., & Room, R.	2020	Restricting alcohol outlet density through cumulative impact provisions in planning law: Challenges and opportunities for local governments	Australia	Qualitative study Target population: implementers (i.e., local officers)	Barriers: limited availability of data for decision making, and insufficient guidelines for implementation	[21]
12.	Wright, A.	2019	Local alcohol policy implementation in Scotland: Understanding the role of accountability within licensing	Scotland	Qualitative study Target population: implementers (i.e., local authorities who implemented licensing policy at local level, and national alcohol policy actors involved in the process of development or delivery of alcohol control policy)	Barriers: lack of accountability of implementing agencies	[22]
	Drink-driving measures						
13.	Eichelberger, A. H., & McCartt, A. T.	2016	Impaired driving enforcement practices among state and local law enforcement agencies in the United States	USA	Quantitative study Target population: implementing agencies (i.e., law enforcement agencies)	Barriers: limited numbers of staff, lack of funding, and excessive paperwork	[23]
14.	Fell, J. C., Ferguson, S. A., Williams, A. F., & Fields, M.	2003	Why are sobriety checkpoints not widely adopted as an enforcement strategy in the United States?	USA	Mixed methods Target population: implementing agencies (i.e., law enforcement agencies)	Facilitators: Organisational support, police manpower, funding, and belief in intervention cost-effectiveness, and public support	[24]
15.	Fiorentino, D. D., & Martin, B. D.	2018	Survey regarding the 0.05 blood alcohol concentration limit for driving in the United States	USA	Mixed-methods Target population: implementers (i.e., law enforcement officers, prosecutors, defence attorneys, and judges)	Barriers: perceived drink-driving measures as a burden (i.e., perceived economic burden of implementing drink-driving measures with BAC level of 0.05)	[25]
16.	Grohosky, A. R., Moore, K. A., & Ochshorn, E.	2007	An alcohol policy evaluation of drinking and driving in Hillsborough County, Florida	USA	Qualitative study Target population: implementing agencies (i.e., enforcement agencies, including police, state attorney, and treatment providers)	Barriers: gaps in existing regulation, heavy workload of key enforcement agencies, and poor communications between enforcement agencies Facilitators: providing education among key stakeholders, raising public awareness and establishing collaboration among stakeholders	[26]
17.	Jia, K., Fleiter, J., King, M., Sheehan, M., Ma, W., Lei, J., & Zhang, J.	2016	Alcohol-related driving in China: Countermeasure implications of research conducted in two cities	China	Mixed-methods Target population: implementers and general drivers	Barriers: insufficient police officers and equipment, insufficient funding, and lack of awareness on the effectiveness of drink driving measures	[27]
	At least two policies						
18.	Abiona, O., Oluwasanu, M., & Oladepo, O.	2019	Analysis of alcohol policy in Nigeria: Multi-sectoral action and the integration of the WHO “best-buy” interventions	Nigeria	Qualitative study Target population: policy actors	Barriers: lack of awareness among policymakers on alcohol-related problems, failure of the government to strengthen systems and structure for alcohol control, lack of funding, poor literacy and deployment of regulatory agencies, no establishment of regulatory agencies, lack of legislation to regulate the alcohol industry, and industry promoted drinking norms	[28]
19.	Casswell, S., Morojele, N., Williams, P. P., Chaiyasong, S., Gordon, R., Gray-Phillip, G., Parry, C. D. H.	2018	The Alcohol Environment Protocol: A new tool for alcohol policy	Multi- country: Scotland, New Zealand, St. Kitts and Nevis, Thailand, South Africa, Vietnam	Mixed-methods Target population: implementers	Barriers: insufficient resources	[29]
20.	Kaewpramkusol, R., Senior, K., Chenhall, R., Nanthamongkolchai, S., & Chaiyasong, S.	2018	Qualitative exploration of Thai alcohol policy in regulating availability and access	Thailand	Qualitative study Target population: implementers (i.e., government officers, academia, and civil society)	Barriers: weak alcohol regulation, lack of community involvement during implementation, conflict of interest (public health versus economic interest), insufficient allocation of resources, and high numbers of alcohol outlets resulting in high workload for monitoring law compliance	[30]
21.	Randerson, S., Casswell, S., & Huckle, T.	2018	Changes in New Zealand’s alcohol environment following implementation of the sale and supply of alcohol act (2012)	New Zealand	Mixed-methods Target population: implementers (i.e., police officers, liquor licensing inspectors, and public health officers)	Barriers: difficulty in gathering sufficient evidence to oppose new licensing, compromises between economic and public health goals, difficulties in enforcement around social supply occurring in a private setting, lack of public concern in social supply, lack of resources/ investment in monitoring data, acceptability of intoxicated behaviours among enforcing officers, insufficient staff, low priority among implementers, and difficulties in assessing intoxication	[31]

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
