# Peer review of "Barriers and Facilitators to the Implementation of Effective Alcohol Control Policies: A Scoping Review"

_ijerph, 2022, doi:10.3390/ijerph19116742_

Round 1

Reviewer 1 Report

The authors present a systematic review on the literature about barriers to implementation of alcohol control policies. They identify five main barriers that occur consistently throughout the literature: resource constraint; legal loopholes; lack of evidence to support policy implementation, low priority of policy implementation; and insufficient skills of implementers. Overall, the authors do a nice job of synthesizing available results on barriers that policy makers face in trying to implement policies aimed at reducing alcohol consumption. I only have some minor comments.

Comments:

In the introduction, the authors seem to conflate use, harmful use, and problems. While reducing overall consumption will likely in turn reduce harmful use, it is important to note that alcohol consumption and alcohol related problems are related, but distinct, phenomena. Policies aimed at reducing consumption may not have large impacts on reducing alcohol related problems if they tend to deter those who are at the lighter end of the alcohol consumption. It is worth noting that policies aimed at reducing alcohol consumption may be different from those that reduce problem or harmful use.

It may also be worth noting that cultural attitudes towards alcohol use ( or cultural attitudes more generally) are difficult to change.

Author Response

Dear reviewer,

We appreciate your comments.  We revised the manuscript regarding your suggestion. Please, see attached file for more explanation for each response.

Best regards,

Jintana,

on behalf of the authors

Reviewer 2 Report

The presented manuscript includes an interesting study that draws on existing literature examining the barriers and facilitators to implementing effective alcohol control policies. It is timely and scientifically sound and properly written, following all the guidelines for publications of scientific articles and it is within the scope of the journal. The abstract of the manuscript is written in great detail and fully covers all parts of the manuscript. The tables and figures additionally facilitate the full evaluation of the content contained in the manuscript and significantly increase its value. The introduction provides an interesting admission to the topic. The methods are described in a detailed and clear manner. In my opinion, the conclusions section should be significantly shortened because it is a certain repetition of the content already contained. However, in my opinion the most serious problem is the novelty of the subject of this manuscript. The Authors point out that this is the first study of this type on the given topic, however, there are already others, e.g. Barriers and facilitators for the implementation of the integrated public policy for alcohol, drug, tobacco, and gambling prevention: a qualitative study. Drugs: Education Prevention and Policy 27 (2): 1-9 DOI: 10.1080 / 09687637.2019.1595527. The Authors should clearly indicate how their manuscript differs from those already existed.

Author Response

(The authors gave the same response as above.)

Reviewer 3 Report

An excellent review- pleasure to read.  My only concern and it may be my copy but I found the main table difficult to follow at time.  If you have not done so I would suggest that this is produced using a windows grid.  If you have already done so please ignore this request.

Author Response

(The authors gave the same response as above.)
